

# Drivers of global irrigation expansion: the role of discrete global grid choice

Sophie Wagner [1], Fabian Stenzel [2], Tobias Krueger [3], and Jana de Wiljes [4,5]

[1]University of Potsdam, August-Bebel-Straße 89, 14482 Potsdam-Griebnitzsee
[2]Potsdam Institute for Climate Impact Research (PIK), Member of the Leibniz Association, P.O. Box 60 12 03, 14412 Potsdam, Germany, stenzel@pik-potsdam.de
[3]Geography Department & IRI THESys, Humboldt-Universität zu Berlin, Unter den Linden 6, 10099 Berlin, tobias.krueger@hu-berlin.de
[4]Institute for Mathematics, Ilmenau University of Technology, Weimarer, 98693 Ilmenau, jana.de-wiljes@tu-ilmenau.de
[5]School of Engineering Sciences, Department of Computational Engineering, Yliopistonkatu 34 53850 Lappeenranta, Finland LUT University, Finland

**Correspondence:** Sophie Wagner (sopwagner@uni-potsdam.de)

**Abstract.** Global statistical irrigation modeling relies on geospatial data and traditionally adopts a discrete global grid based on longitude-latitude reference. However, this system introduces area distortion, which may lead to biased results. We propose using the ISEA3H geodesic grid based on hexagonal cells, enabling efficient and distortion-free representation of spherical data. To understand the impact of discrete global grid choice, we employ a non-parametric statistical framework, utilizing
random forest methods, to identify main drivers of historical global irrigation expansion amongst others, also using outputs from the global dynamic vegetation model LPJmL.

Irrigation is critical for food security amidst growing population, changing consumption patterns, and climate change. It significantly boosts crop yields but also alters the natural water cycle and global water resources. Understanding past irrigation expansion and its drivers is vital for global change research, resource assessment, and predicting future trends.

We compare the predictive accuracy, the simulated irrigation patterns and identification of irrigation drivers between the two grid choices. Results demonstrate that using the ISEA3H geodesic grid increases the predictive accuracy by 29% compared to the longitude-latitude grid. The model identifies population density, potential productivity increase, evaporation, precipitation, and water discharge as key drivers of historical global irrigation expansion. GDP per capita also shows minimal influence.

We conclude that the geodesic discrete global grid significantly affects predicted irrigation patterns and identification of
drivers, and thus has the potential to enhance statistical modeling, which warrants further exploration in future research across related fields. This analysis lays the foundation for comprehending historical global irrigation expansion.

## 1 Introduction

About 80% of data being produced is of geospatial nature (Hahmann and Burghardt, 2013). While the construction of maps and the referencing of locations on the Earth's surface has a very long history, it is becoming increasingly important to find efficient
ways to process, integrate and analyze geospatial data to solve problems in times of globalization. To that end, geographic grid





systems are used to project the geographic space into a mathematical space where algorithms and statistical methods can be applied.

The most widely used grid system is the geographic coordinate system (using latitude and longitude lines), which dates back to the third-century BCE (McPhail, 2011; Ware et al., 2020). A great advantage of this system is that it can be stored compactly

and used easily for computations (Ware et al., 2020). Unfortunately, when portrayed on a sphere, grids based on geographic coordinates suffer from cell area distortion due to the converging lines of equal longitude. In the context of global statistical modelling, this ultimately results in oversampling of the northernmost regions.

A great number of alternative Discrete Global Grid Systems (DGGS) have emerged to fulfill the needs of different research fields and modelling strategies. Specific criteria for what comprises a DGGS have evolved over time, starting from Goodchild's

critera (Goodchild, 1994). Today, standard Earth grid systems are documented by the Open Geospatial Consortium (OGC) and the International Organization for Standardization (ISO) (ISO, 2021; Purss, 2015).

In this paper, we propose to use a global DGGS based on a hexagonal tessellation of the Earth's surface in the context of modelling global historical irrigation expansion. This grid was introduced by Sahr et al. (2003) and has gained large popularity in many research contexts. Two recent examples of open-source DGGS libraries that are based on hexagonal grid structures are

the H3 system, developed by Uber (2022) and DGGRID (Barnes and Sahr, 2017). Mechenich and Zliobaite (2023) recently presented the Eco-ISEA3H database that consists of global spatial data characterizing climate, geology, land cover, physical and human geography, and the geographic ranges of nearly 900 large mammalian species. In contrast to grid cells induced by the longitude-latitude graticule, hexagonal cells are able to cover almost the entire surface of the Earth without suffering from area distortion. That way, all regions have the same influence in the statistical model.

Recently, hexagonal mesh grids have gained popularity among hydrology researchers (Li et al., 2022). A group of hydrological functions on hexagonal meshes, such as flow direction and accumulation, stream networks, or watershed boundary extraction, were explored by Liao et al. (2020). The authors show that their algorithm's performance is better when considering the hexagonal-mesh-based output compared to the traditional square-mesh-based output. Wang et al. (2020) study valley networks and model valley lines based on hexagonal grids. Compared to traditional square grids, the study shows that using

the hexagonal grid leads to a higher location accuracy. In another study, Wright (2019) develops a regular hierarchical surface model where hydrological computation was generalized on hexagonal and triangular grids. Additionally, there has been an increasing interest in managing geospatial data and developing models to solve real-world problems by using the open-source DGGRID library (Hojati and Robertson, 2020; Li et al., 2021; Chaudhuri et al., 2021; Robertson et al., 2020; Li et al., 2022). To the best of our knowledge, we are the first to utilize the *Icosahedral Equal Area aperture 3 Hexagon geodesic Discrete*

*Global Grid System* (ISEA3H) DGGS in the context of building a statistical model for historical global irrigation expansion.

The second aim of this paper is to contribute to the literature on global irrigation expansion. Irrigation is crucial to ensure the world's food security. A growing human population, shifting consumption patterns and climate change increase the pressure on agricultural production (Foley et al., 2011). To meet the growing human food demand, irrigation has rapidly increased over the last century as it increases crop yields (Siebert et al., 2015). In the year 2000, approximately $40\%$ of the global food production

was harvested on irrigated land, utilizing only 20% of the total farming area (Schultz et al., 2005). To achieve this agricultural





intensification, a large amount of fresh water is needed. Consequently, irrigation alters the hydrological cycle significantly (Zohaib and Choi, 2020). At a global scale, irrigation is responsible for about 60% of total fresh water withdrawals and 80% of total fresh water consumption (Döll et al., 2014; Siebert et al., 2015). It is therefore important to understand the past evolution of irrigation expansion and its main drivers for global change research, the assessment of resources and for predicting future
developments.

There have been a few studies on the drivers of global irrigation in previous years. Neumann et al. (2011) investigate the global irrigation pattern in the year 2000. Using a multilevel approach, they model irrigation as a function of biophysical and socioeconomic factors. Their results show that biophysical factors have significant influence on irrigation. Additionally, the authors provide suggestive evidence that socioeconomic factors play a role for irrigation. However, it is emphasised that the
model suffers from uncertainty due to the lack of spatially explicit socioeconomic information and the possibility of external influences, such as public investments. While our model also faces these limitations, we are able to extend the analysis by including a historical dimension.

Puy et al. (2020) investigate uncertainties in published projections of global irrigation expansion for the year 2050. By comparing different projected estimates of irrigated area to a simple model predicting irrigated area as a function of only
population size, constrained by water and land availability, taking into account parametric and model uncertainties, the authors postulate, that current models underestimate future irrigated areas.

Further, recent studies attempt to develop global irrigation maps, mostly using a combination of remote sensing, machine learning methods and climate data, to enhance knowledge about current irrigated areas (Meier et al., 2018; Salmon et al., 2015; Nagaraj et al., 2021).
We contribute to the literature on global irrigation expansion by investigating the drivers of the historical expansion between 1902 and 2000, using a novel method to statistically model global irrigation. In that, we distinguish between the factors that influence the probability of a grid cell being irrigated, i.e. the decision to irrigate instead of remaining rainfed, and the irrigation intensity, once a grid cell is irrigated.

Our results show, that using an ISEA3H grid based on hexagonal grid cells, instead of the common longitude-latitude grid,
leads to an increase in mapping accuracy of 28%. Overall, our results show that population density has the strongest influence on the decision to irrigate. We also find that the median increase in potential productivity and the predictors related to water availability play a role for the likelihood of irrigation occurring. Considering the drivers of the amount of irrigation, we see that population density has a large positive impact. In addition evaporation, discharge, and the median increase in potential productivity have a positive influence on the irrigation amount. Precipitation influences the irrigation amount negatively. We
also find that GDP per capita plays a role for the amount or irrigation used in a grid cell.

## 2 Data

Our objective in this study is to analyze the choice of discrete global grid system when modelling the historical evolution of global irrigation expansion. Therefore, we use a study area that covers all of the global land surface, excluding Antarctica.





Furthermore, we consider data between 1902 and 2005 to be able to fully capture the historical evolution of irrigation expansion

of the last century.

Our analysis builds on a data set that consists of a simulation output from the *Lund-Potsdam-Jena managed Land* (LPJml) model (Sitch et al., 2003; Bondeau et al., 2007) and historical economic data from the *Maddison Project Database* (Inklaar et al., 2018).

LPJmL is a process-based dynamical global vegetation, hydrology, and crop model simulating natural and managed vege-

tation growth based on soil, climate, and management input at a daily resolution and at a global $0.5° \times 0.5°$ spatial grid scale, resulting in a total amount of 67420 terrestrial grid cells per time unit in each variable (Schaphoff et al., 2018).

We prescribe an agricultural land use data set based on the *History Database of the Global Environment* (HYDE) (Klein Goldewijk et al., 2017) with additional assumptions on irrigation systems and extent of areas equipped for irrigation by Jägermeyr et al. (2015) and based on the *Global Historical Irrigation Data Set* (HID). One advantage of the HID is, that the evolution

of land irrigation was implemented using official land use data and is therefore independent of socioeconomic information, such as gross domestic product or population density (Siebert et al., 2015). Hence, the relationship of irrigation and socioeconomic variables can safely be analyzed. As climate input, the *Climatic Research Unit Timeseries* (Harris et al., 2014) is used. Whether a crop actually needs irrigation is internally decided by the LPJmL simulation based on biophysical constraints, and constrained by surface water availability (Schaphoff et al., 2018).

From the simulation, we obtain the direct output variables precipitation, evaporation, discharge, crop yield, and the actually irrigated fraction for each grid cell. Additionally the median potential increase in crop yield productivity is derived. It is estimated from two separate synthetic simulations, where potential yields for each crop and grid cell are compared with and without irrigation. The variables are summarized to yearly estimates for each grid cell to obtain a time series for the years 1901 to 2005.

To complement the LPJmL data, we use the Maddison Project database on the historical performance of the world economy (Inklaar et al., 2018; Bolt and Zanden, 2014). Of particular interest is the gross domestic product (GDP) per capita time series, consisting of estimates of comparative levels of real GDP per capita in recent time periods, combined with long-term time series growth of GDP per capita. Even though the Maddison Project database yields state of the art historical economic data, there are many countries without an estimation of GDP per capita in the time period 1900 to 1960, leading to missing data.

## 2.1 Variables

As dependent variable we use the fraction of a grid cell that is actually irrigated. If the area of a grid cell is fully irrigated, the variable will have the value "1", if none of the area is irrigated, the variable will have the value "0". Note, since the grid cells in the standard longitude-latitude grid change area in proportion to latitude, the values of irrigation fraction between grid cells can not be directly compared. The global irrigation fraction map, based on HID data from 2000, is displayed in Figure A4.

The selection of potential drivers of irrigation expansion was led by existing literature and data availability (see Table 1). We consider the following variables for explaining irrigation fraction: population density, precipitation, discharge, evaporation, potential yield increase through irrigation, and GDP per capita.



The GDP per capita data is available at national level and broken down to a $0.5° \times 0.5°$ grid scale, by assigning the country's value to all grid cells in a country. Since there are observations missing, especially in the earlier time periods, the variable is split up into the categories "high income", "upper middle income", "lower middle income", "low income" and "missing", following the methodology of Hastie et al. (2009) and the World Bank's classification of GDP per capita from 2011 (World Bank, 2011). The classification can be found in the supplementary material (Table A2). That way, we treat the missing values as an additional category and are able to include all observations in our analysis.

We report the pairwise Pearson correlation coefficients (supplementary material, Table A3) and variance inflation factors (supplementary material, Table A4) to investigate multicollinearity between the continuous predictor variables, following the methodology in Rufin et al. (2018). We find that all pairwise Pearson correlation coefficients are below the threshold of 0.7 (Dormann et al., 2013), except for the pair "Precipitation" and "Evaporation", where we find a value of 0.72. The variance inflation factors are below the tight threshold of 5, indicating that the predictor variables are sufficiently independent for our analysis (James et al., 2013).

Table 1: Potential predictors and hypotheses

| Predictor Variable | Hypothesis | Supporting Literature |
|---|---|---|
| Precipitation (mm/year) | Irrigation requirements increase in cropland regions where precipitation levels are declining. | (Neumann et al. (2011)), (Döll and Siebert (2002)), (Siebert et al. (2015)) |
| Discharge ($hm^3$/year) | Surface water availability allows for irrigation water withdrawals. | (Neumann et al. (2011)), (Gerten et al. (2008)) |
| Evaporation (mm/year) | High evaporation leads to an increasing demand of water and therefore increases the probability of irrigation. | (Neumann et al. (2011)), (Rufin et al. (2018)) |
| Median Increase in Productivity (% of $\Delta$ gC/$m^2$) | If the potential increase in agricultural productivity is large, the corresponding area is more likely to receive irrigation. | (FAO and of the United Nations (2011)), (Sauer et al. (2010)) |
| Population Density (cap/$m^2$) | Intensive irrigation occurs under high population densities. The rapidly growing world population increases the demand for food and, therefore, leads to an expansion or intensification of agriculture globally but also around high-density centres. | (Neumann et al. (2011)), (Rufin et al. (2018)), (Boretti and Rosa (2019)), (Sauer et al. (2010)) |





| Predictor Variable | Hypothesis | Supporting Literature |
|---|---|---|
| GDP ($ US /cap) | A high GDP per capita leads to a higher probability of irrigation, since farmers can afford irrigation systems or are more likely to receive subsidies. GDP is also highly correlated with government effectiveness and hence serves as a proxy. A high national government effectiveness strengthens irrigation infrastructure. | (Neumann et al. (2011)), (Rufin et al. (2018)), (Boretti and Rosa (2019)), (Sauer et al. (2010)) |

## 2.2 Descriptive Statistics

Overall, irrigated area is expanding over the whole study period. The share of grid cells in which irrigation is observed increases from about 10% in 1902 to about 31% in 2005. The largest increase in irrigated area can be seen in southeastern Asia, Middle and South America, central America and eastern Asia. These statistics are in accordance with the findings by Siebert et al. (2015), who investigate areas equipped for irrigation.

Even though irrigated land is expanding, the data is highly imbalanced: looking at the whole study period, about 75% of the observed irrigation fractions are zero, whereas only about 25% are non-zero. Figure A2 in the appendix shows the histogram of the irrigation fraction.

The summary of the overall descriptive statistics of irrigation fraction and the potential predictors can be found in Table A1. The dependent variable, irrigation fraction, ranges from zero to 0.922 with a mean of 0.008 in the longitude-latitude grid.

The global temporal evolution of the predictor variables is illustrated in Figure A1 in the appendix. The global mean evaporation is increasing over the last century as well as the GDP per capita. We also see a slightly increasing trend of the global amount of precipitation. For the remaining variables, there is no clear detectable trend in the global mean. However, it is expected that there are local trends that are not captured in the global mean values.

## 3 Method

### 3.1 Spatial Resolution

The latitude-longitude projection yields a world map which appeals to the human eye for its plane appearance but also faces some limitations. The grid cells that are induced by the longitude-latitude graticule are not of the same area. One degree of latitude represents the same horizontal distance anywhere on the Earth's surface. However, because lines of equal longitude are farthest apart at the equator and converge to single points at the geographic poles, the horizontal distance equivalent to one degree of longitude, varies with latitude (Budic et al., 2016). For statistical analysis, this means that regions nearer the poles, which are smaller in area yet weighted the same as larger areas nearer the equator, overly contribute to models and have therefore a higher influence on the results. For land-based analysis such as ours this is particularly relevant for the northern hemisphere were a considerable land mass is located nearer the poles.

The disadvantages of discrete global grids based on the geographic coordinate system led to a number of alternatives. One first idea, could be to weight each grid cell by its area and therefore its relative importance for the statistical model. A more





direct approach would be to use a discrete grid system that subdivides the Earth's surface into equally sized grid cells, allowing for an efficient identification of patterns, trends and relationships across different geographic scales.

Sahr et al. (2003) introduced a class of reference grids based on convex regular polyhedra, called *geodesic Discrete Global Grid Systems* (geodesic DGGS). The underlying idea is to use the topological equivalence of regular polyhedra and the sphere.

Based on five design choices, the resulting grid partitions the Earth into equally-sized cells. The first choice involves picking a base polyhedron. The distortion of area tends to be smaller the smaller the faces of the base polyhedron (Sahr et al., 2003). Therefore, in this study we choose the icosahedron as a starting point, as it has the smallest face sizes compared to the other regular polyhedra. The second design choice requires to pick a method of partitioning the surface of the icosahedron. Hexagons have been found in many research fields to be the optimal choice for discrete gridding and location representation (Apte et al.,

2013; Uher et al., 2019). One unique property of a hexagonal grid is its uniform adjacency; each cell in a hexagonal grid has six neighbors, all of which share an edge with the cell, and all of which have centers exactly the same distance away from their neighbouring cells.

Thirdly, one has to decide on the orientation of the base icosahedron relative to the Earth's surface. In other words, it is required to choose the location of the pentagonal cells, as they are located at the vertices of the icosahedron. The most common

choice is to place the pentagons, such that only one is centered on land (Sahr et al., 2015). This specific orientation is also symmetric about the equator.

In a fourth step, a method for the transformation between the surface of the Earth and the surface of the icosahedron, upon which the hexagonal grid is constructed has to be selected. Our choice is the only known equal-area icosahedral geodesic DGG projection, called the Snyder Icosahedral Equal Area (ISEA) projection (Snyder, 1992).

Lastly, a recursive partitioning method must be picked in order to create different spatial resolutions. Such method is characterized by the ratio of cell areas at a given grid resolution and the next coarser resolution. This ratio is called *aperture*. We will consider aperture 3 hexagonal grid cells, meaning that the increase of the resolution by one, leads to grid cells with an area of a third of the original cell area. Figure A5 in the appendix illustrates the partitioning method.

After making these five basic construction choices, the result can be referred to as an *Icosahedral Snyder Equal Area aperture*

*3 Hexagon geodesic Discrete Global Grid System* (ISEA3H DGGS).

## 3.2 Data Transformation

Our data set is originally structured upon the standard longitude-latitude reference system at a $0.5° \times 0.5°$ spatial resolution, where the location of a grid cell is accessible through the latitude and longitude of its center point. This leaves us with a number of 67,420 land grid cells per year. To understand the role of the discrete global grid choice and compare between the

standard approach and the distortion-free geodesic alternative, we construct a geodesic DGG reference frame and transform our observations accordingly.

Using the freely available *R* package *dggridR* provided by Barnes and Sahr (2017), we construct an ISEA3H discrete global grid at resolution 7. This translates to a hexagonal grid, where the grid cell centers are 160 km apart from each other. After the transformation, we have a number of 7,383 grid cells per year.





The original data are projected into the hexagonal cells. Depending on the degree of latitude, different amounts of cell centers of the original grid end up in each hexagon. The center counts pattern is visualized in Figure 1.

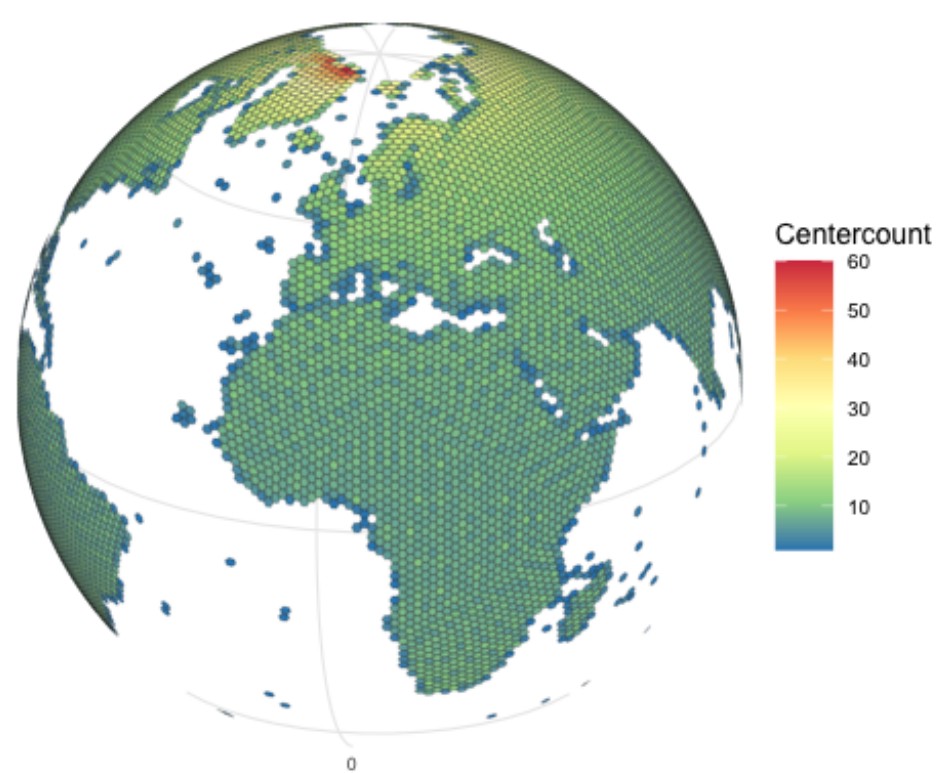

**Figure 1.** Number of grid cell centers of the longitude-latitude grid that fall into each hexagonal grid cell of the resolution 7 ISEA3H grid. The color pattern shows that in the northern (and southern) parts, more grid cell centers fall into each hexagonal cells compared to the areas around the equator.

After mapping the original cell centers into the ISEA3H grid, the mean of all observations within each hexagonal cell is taken as the new value in the transformed data set.

### 3.3   Random Forest

We model the observed variation of irrigation fraction with a set of biophysical and socioeconomic predictor variables using a random forest framework.





A random forest consists of a set of individual decision trees that operate as an ensemble. The method was introduced by Breiman (2001) and is nowadays a widely used machine learning technique, because it tends to have high prediction power with little tuning of its parameters. A random forest captures non-linearity, is able to deal with imbalanced data and estimates of variable importance are readily available (Strobl et al., 2009).

Depending on the response variable, the decision trees of the random forest perform either classification or regression, based on a recursive partitioning method. At each step, a decision tree finds the optimal split that minimises "impurity", until a specific stopping criterion is met. Impurity is a measure of the homogeneity of the class labels at a certain node in the decision tree. There are several different ways to define the impurity measure. Following Wright and Ziegler (2017), we use the estimated response variance for regression trees and the Gini-index for classification trees as measures for impurity. Please find the precise definitions in Algorithm 1. Ultimately, the recursive partitioning method repeatedly splits the data into potentially high-dimensional rectangular partitions of the predictor space, choosing those for which the response data are relatively homogeneous (Strobl et al., 2009).

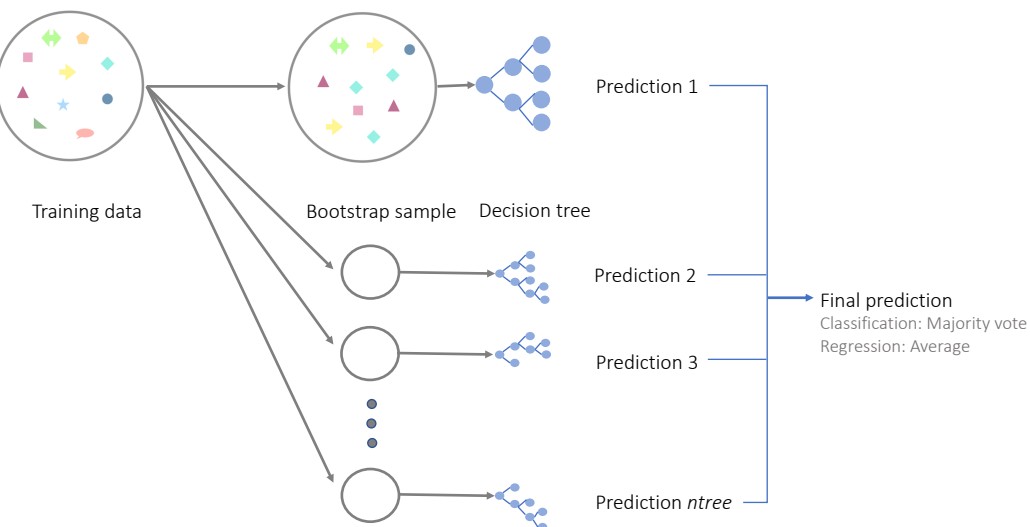

**Figure 2.** Synthetic representation of the random forest method.

Usually, a random forest consists of several hundred or thousands of trees and combines the results of their predictions (Strobl et al., 2009). The trees are built on bootstrapped samples of the training data. On average, each bootstrap sample contains 63.2% unique observations (Breiman, 2001), which are called in-bag samples. Samples not selected are called out-of-bag (OOB) samples and are used to estimate the prediction accuracy, also called *OOB error*. These error estimates provide



an accurate measurement of the generalization error as they are similar to the results obtained through $K$-fold cross-validation (Wolpert and Macready, 1996). However, the OOB error can be sensitive to the number of random predictors used at each split

($mtry$) and the number of trees ($ntree$) in the random forest (Huang and Boutros, 2016). Generally, the accuracy is increasing as the number of trees increases. However, literature has shown that the accuracy levels at a certain number of trees, depending on the specific learning task (Oshiro et al., 2012). The parameter $mtry$ has been found to have a high influence on prediction accuracy and should be selected carefully (Huang and Boutros, 2016; Bernard et al., 2009; Probst et al., 2019). We focus on $ntree$ and $mtry$ as tuning parameters in our model set-up, to achieve a high performing random forest model. Figure 2

illustrates the random forest framework.

The step-by-step process of building a classification and regression random forest follows Algorithm 1. To cope with the

---

**Algorithm 1** Random Forest

Given a data set $\{(x_i, y_i) : i = 1, ..., n\}$, where $y_i$ is the $i$th observed dependent variable and $x_i = (X_1, ..., X_p)$ is a $p$-dimensional predictor vector.

**Step 1.** Draw a number of $ntree$ bootstrap samples sets from the training data set. Each sample is the same size as the training data set. The number $ntree$ is a tuning parameter, also referred to as the number of trees in the forest.

**Step 2.** At each node split a random number of $mtry$ predictors out of all $P$ predictors are considered, i.e. $X_i, i = 1, ..., mtry$ with $mtry < P$. The number $mtry$ is another tuning parameter.

**Step 3.** Predictor $j$ splits the observations $\{y_i\}, i = 1, ..., n$ into the most uniform binary regions $R_l := \{X | X_j \leq c\}$ and $R_r := \{X | X_j > c\}$ according to the following impurity measures:

  – (Regression) weighted residual sums of squares

$$min_{j,c}\left(p(R_l) \sum_{j:y_j \in R_l} (y_j - \bar{y}_{R_l})^2 + p(R_r) \sum_{j:y_j \in R_r} (y_j - \bar{y}_{R_r})^2\right), \tag{1}$$

  where $\bar{y}_{R_l}$ and $n_l$ are the mean and number of observations in region $R_l$, $\bar{y}_{R_r}$ and $n_r$ are the mean and number of observations in region $R_r$ and $p(R_k) = n_k/n$ is the proportion of observations in Region $k \in \{l, r\}$.

  – (Classification) Gini impurity

$$min_{j,c}\left(n_l \hat{p}_l(1 - \hat{p}_l) + n_r \hat{p}_r(1 - \hat{p}_r)\right), \tag{2}$$

  where $\hat{p}_k$ is the proportion of sample points that were sent to node $k \in \{l, r\}$ from the previous node.

**Step 4.** Repeat steps 2-3 until each terminal node reaches the predefined minimum number of observations $min.node.size$.

**Output.** The algorithm forms a partition of the data into $M$ regions $R_1, ..., R_M$, and model the response as a constant $r_m$, i.e.:

$$f_{RF}(x) = \sum_{m=1}^{M} r_m I(x \in R_m). \tag{3}$$

---





imbalance of our dependent variable, two random forests are trained to build a hurdle model. A classification random forest is trained to predict whether a grid cell is irrigated or not and a regression random forest is trained to predict the magnitude of irrigation. The idea is, to use both models to build a stacked final model that predicts irrigation fraction based on the available

predictors. That way, we account for the zero-inflated distribution of irrigation fraction.

We use the freely available $R$ package *ranger*, developed by Wright and Ziegler (2017) for the training and validation of the random forests.

### 3.3.1  Parameter Tuning and Model Setup

We use cross-validation (CV) to tune our random forest models and find the values of the parameters *ntree*, i.e. the number of

decision trees, and *mtry*, i.e. the number of predictors to be considered at each split, that yield the highest predictive accuracy. Data from the years 1902 to 1999 are taken as the training sample, observations from the years 2001 to 2005 are used as the validation sample and the remaining data from 2000 serves as a test sample. Due to computational constraints, we apply a sub-sampling routine in order to find our model parameter values in a reasonable amount of time. For the classification random forests, a balanced sample of 10% is drawn in each CV fold, with 50% irrigated and 50% rainfed grid cells, using random over-

and under-sampling methods from the $R$ package ROSE, made available by Lunardon et al. (2014). For the regression random forest, all irrigated grid cells are used for training.

As accuracy measures, we take the OOB error and the validation error. We set the minimal number of data points at each terminal node, i.e. *min.node.size* equal to 10, which serves as a stopping criterion. For the parameter *ntree*, we consider the values 50, 300, 500, 800, 1000, 2000, 3000, 4000 and 5000 and for *mtry* we try all values between 1 and 5 at 0.5 increments

for both the classification and the regression random forest. We perform 50-fold CV to train the classification and regression random forests for both grid choices separately.

The resulting accuracy for each forest and each tuning parameter value can be seen in Figures A6 and A7. Taking the OOB error and the validation error into account, we choose $ntree = 1000$ and $mtry = 1.5$ for the classification random forest and $ntree = 4000$ and $mtry = 5$ for the regression random forest for the longitude-latitude grid. In the ISEA3H grid we set

$ntree = 1000$ and $mtry = 5$ for the classification random forest and $ntree = 4000$ and $mtry = 5$ for the regression random forest.

After setting the tuning parameters according to the optimal values, the stacked random forest model can be used to produce predictions. We use the test data to evaluate the prediction accuracy. The final prediction of the model is obtained by multiplying the prediction of the classification random forest and prediction of the regression random forest. We compare the final prediction

results for both grid choices.




## 4 Results

### 4.1 Grid choice

We compare the longitude-latitude grid with the ISEA3H grid based on their predictive power and their ability to identify the drivers of the global irrigation expansion. To get a first intuition about differences in predictive power of the two model setups,

we create a binned scatterplot of the predicted irrigation fraction of the test data against the actual values of irrigation fraction for both grids. In that way, the 45 degree line mechanically indicates correctly predicted irrigation fraction values. Figure 3(a) shows the result for the longitude-latitude grid model and Figure 3(b) displays the result for the ISEA3H grid model. The comparison suggests that the ISEA3H grid model has a higher prediction accuracy, since the point values scatter more closely around the 45 degree line.

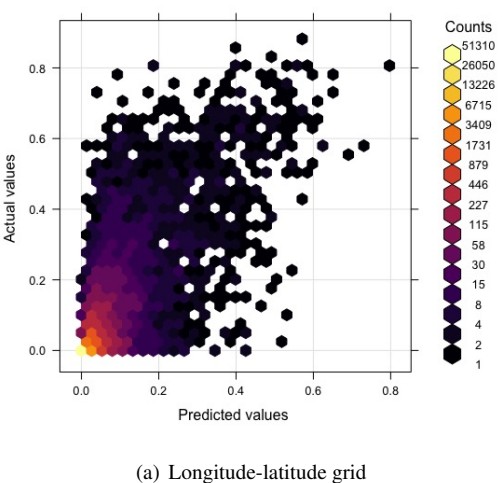
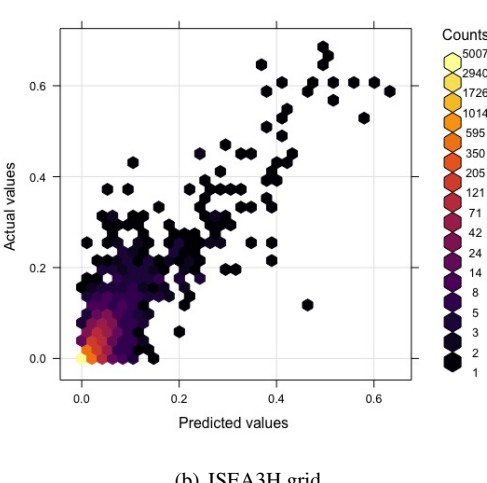

(a) Longitude-latitude grid

(b) ISEA3H grid

**Figure 3.** Binned scatter plot of predicted vs. actual irrigation fraction values for (a) the longitude-latitude grid and (b) the ISEA3H grid. The prediction is based on the test data.

To further evaluate the difference in predictive accuracy between the two grid choices, we compute the root mean square error (RMSE) and the normalized root mean square error (NRMSE). The RMSE and the NRMSE indices are calculated as

$$RMSE = \sqrt{1/n \sum_{i=1}^{n} (y_i - \hat{y_i})^2} \tag{4}$$

and

$$NRMSE = \frac{RMSE}{sd(y)}, \tag{5}$$

where $y_i$ is the actual value, $\hat{y_i}$ the prediction and $sd(y)$ the standard deviation over all actual values. The RMSE and NRMSE were calculated for the prediction on the test data and compared between grid choices. We additionally evaluate the NRMSE





for both grid choices, after restricting the sample to observations with non-zero irrigation. The results are reported in Table 2. The model with the lower NRMSE is considered the better choice to model irrigation fraction.

**Table 2.** Normalized root mean square error comparison between the longitude-latitude and the ISEA3H grid choice

| | Longitude-Latitude grid | ISEA3H grid | Reduction in NRMSE (%) |
|---|---|---|---|
| | (1) | (2) | (3) |
| **A. All observations** | | | |
| Mean | 0.0156 | 0.0168 | |
| SD | 0.0604 | 0.0525 | |
| RMSE normalized with: | | | |
| SD | 0.676 | 0.484 | 28 |
| Mean | 2.618 | 1.508 | 42 |
| Max-Min | 0.047 | 0.037 | 21 |
| **B. Non-zero observations** | | | |
| Mean | 0.0507 | 0.0337 | |
| SD | 0.1005 | 0.0703 | |
| RMSE normalized with: | | | |
| SD | 0.702 | 0.503 | 29 |
| Mean | 1.390 | 1.05 | 24 |
| Max-Min | 0.081 | 0.052 | 36 |

Notes: Column (1) shows the mean, the standard deviation of the irrigation fraction and the NRMSE values of the longitude-latitude grid choice. Column (2) provides the same for the ISEA3H grid choice. In column (3) the reduction in NRMSE is documented in percent and in comparison to the longitude-latitude grid. Panel A. includes all observations and gives the overall NRMSE estimates. In Panel B. only irrigated areas are included. The NRMSE values here provide insight to how the models for both grid choices perform on actually irrigated terrain.

Generally, we see that lower errors are observed when using the ISEA3H grid. We see a 28% reduction in the NRMSE when using the ISEA3H grid (NRMSE = 0.484) compared to the longitude-latitude grid (NRMSE = 0.676). Considering only irrigated areas, we see a that the ISEA3H grid (NRMSE = 0.503) corresponds to a 29% lower NRMSE compared to the longitude-latitude grid (NRMSE = 0.702).

To check the robustness of our result, we calculate the NRMSE by using the mean and the distance between the minimum and the maximum value as standardizing measures. The results are also reported in Table 2. The general result remains the same, in that we observe lower NRMSE when using the ISEA3H grid.

In a next step, we consider the predicted irrigation fraction pattern. In that, we aim to evaluate how accurately the models predict high and low values of irrigation fraction across the global map. Figure 4 shows the difference between the predicted irrigation fraction pattern and the actual irrigation fraction pattern for (a) the longitude-latitude grid and (b) the ISEA3H grid. The computation is based on the test data. The color scale indicates, if the model predicts the irrigation fraction accurately or suffers from under- or over-prediction. Yellow areas are correctly predicted by the model, orange to red areas correspond to under-prediction and green to blue areas indicate over-predicted irrigation fraction values. Considering the longitude-latitude grid, we see that, irrigation is under-predicted in some areas in India and East Asia and also in few areas in North and South America and Europe. Except for very few parts in India, Central-Africa and North America, we do not see any over-prediction



of irrigation. Looking at the ISEA3H grid, we find the same areas in India and East Asia are slightly over-predicted as well

as some areas in the United States and Europe. Only few areas are under-predicted in India, East-Asia, and Central-Africa. Comparing both grids, we find that the ISEA3H grid is closer to the benchmark in all areas. Especially, the highly irrigated areas in east Asia are better captured by the ISEA3H grid model and we also see less over-prediction in European areas. The maps indicates that the ISEA3H grid is the better choice in predicting the global irrigation fraction pattern.

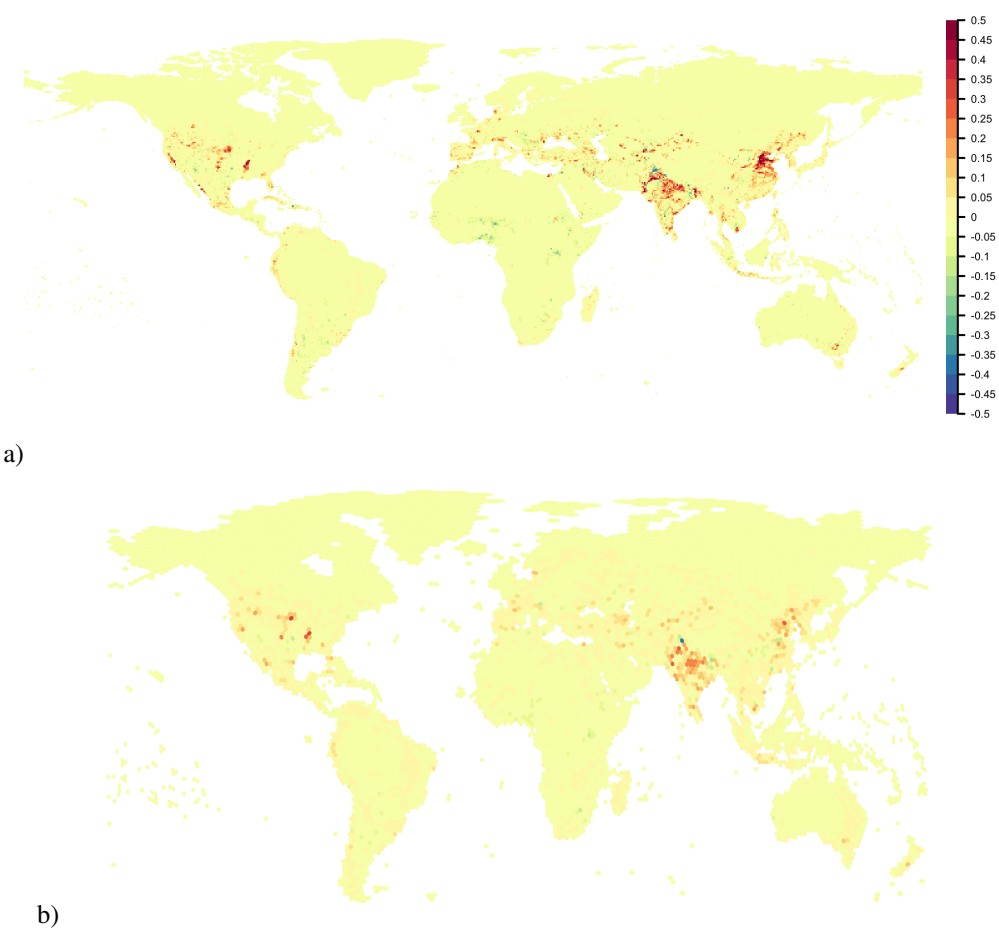

**Figure 4.** Deviation of the predicted irrigation fraction from the observed irrigation fraction in (a) the longitude-latitude grid representation and (b) the ISEA3H grid representation. Green and blue areas indicate an under-prediction of the irrigation value and orange and red values over-prediction. Yellow areas correspond to areas where irrigation values were predicted correctly. The prediction is based on the test data.





## 4.2 Drivers of irrigation expansion

### 4.2.1 Variable importance

We estimate the relative importance of the predictor variables during the random forest estimations. We report the importance of the predictors for the classification random forests, i.e. the probability that irrigation occurs, and the regression random forests, i.e. the irrigation intensity, given that the area is irrigated. The relative importance in the classification random forests are measured in terms of Gini gain and the relative importance for the regression random forests are captured in estimated
response variance. The results are displayed in Figure 5. Panels a) and b) show the results for the longitude-latitude grid choice and panels c) and d) display the results for the ISEA3H grid choice.

The most important driver for the likelihood that an area is irrigated is population density. This is the case for both grid

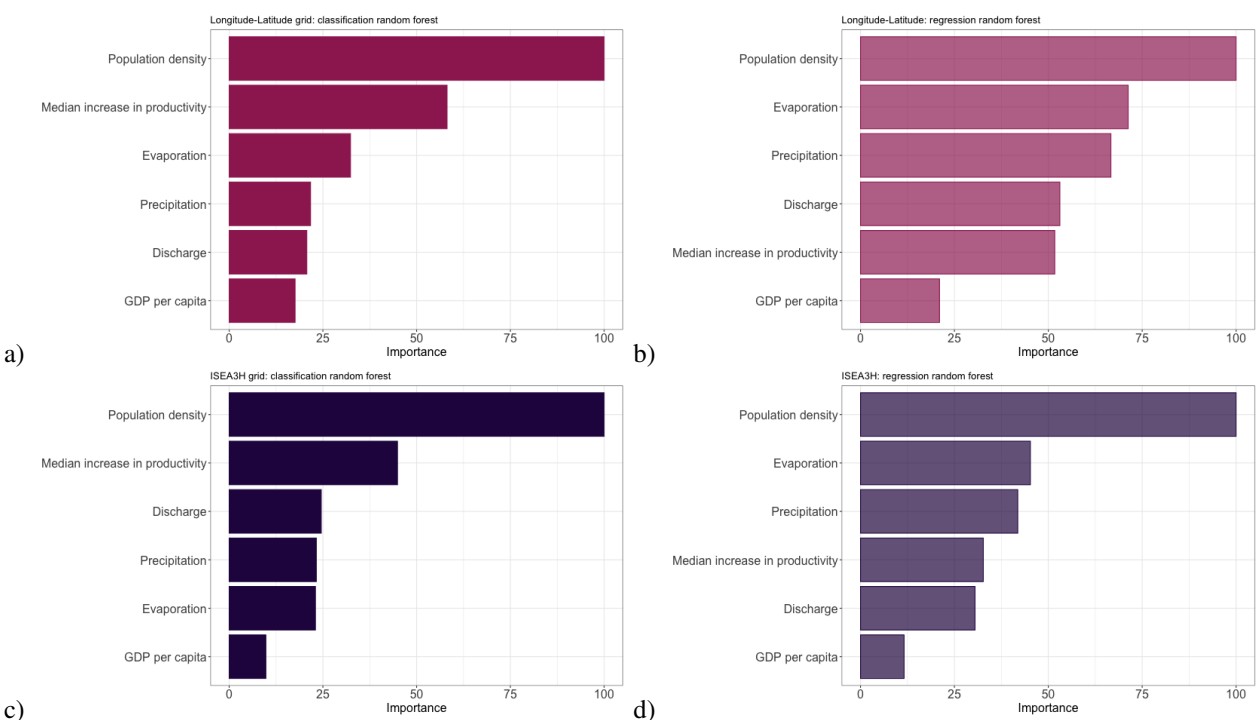

**Figure 5.** Relative importance, measured as decrease of node impurity. The results for the longitude-latitude grid choice can be seen in purple and the results for the ISEA3H grid choice is displayed in magenta. Figure (a) corresponds to the classification and Figure (b) to the regression random forest. Figure (c) corresponds to the classification and (d) to the regression random forest.

choices. The second most important driver is the median potential increase in productivity in terms of crop yield. Evaporation, precipitation and discharge all have a similar influence on the irrigation probability. However, the order of importance is
reversed between the two grid choices. The GDP per capita only has small influence on the decision to irrigate.





The most important driver for the irrigation intensity, given that an area is already irrigated, is also population density. This is followed by evaporation, precipitation, discharge and the median increase in potential productivity, where the order of discharge and the median potential productivity increase is the reversed for the ISEA3H grid choice. The last most important driver is again the GDP per capita, though still having some influence on the models' performance in both grids.

### 4.2.2 Partial dependence

We compute the partial dependence of each predictor variable for both grid choices. The partial dependence is obtained by gradually changing the value of one predictor variable and predicting the outcome variable at each step, while leaving the remaining predictors constant. That way, the functional relationship between the predictor and the dependent variable becomes visible. The larger the value range at the y-axis, the larger the influence of the predictor on the dependent variable. Figure 6
illustrates the results.

Panel a) of Figure 6 illustrates the partial dependence of the predictor variables on the probability of irrigation. Overall, we see very intuitive dependence patterns.

Population density has a positive influence on the probability to irrigate, where the probability sharply increases at the beginning of the population density distribution. In other words, greater population density correlates with an increased likelihood
of irrigation, indicating that metropolitan regions with higher population densities and improved market accessibility are more likely to engage in irrigation. This heightened probability is likely attributed to the requirement of capital investment for establishing irrigation systems. This aligns with the paper by Neumann et al. (2011), who also find a positive association between irrigation and population density.

A similar pattern can be seen for the median potential increase in productivity, the second most influential predictor. This
positive correlation demonstrates that farmers assess the potential increase in crop yield when they decide to implement irrigation systems.

Evaporation also has a positive, almost linearly increasing influence on the irrigation probability. Considering precipitation, our results show that the probability to irrigate decreases with the amount of precipitation until the probability levels and does not change anymore with increasing precipitation values. The amount of available discharge has a negative relationship
to the probability to irrigate for both grid choices at the beginning of the distribution. Looking at the longitude-latitude grid, this changes into a positive correlation, leaving us with a u-shaped dependence curve. Looking at the ISEA3H grid choice, the irrigation probability does not change anymore after reaching a certain discharge level. Overall, these results show that water availability and climatic conditions play a role for the decision to irrigate, leaving rather dry areas and areas with higher evaporation levels more likely to be or become irrigated. Discharge is an accumulated variable of local runoff, with
very high differences between upstream and downstream cells in a watershed. This means that regions with relatively high topography and thus potentially lower degrees of agriculture and irrigation are all coinciding with low discharge values, while the major irrigation areas (India, Pakistan, US, East-Asia, Egypt, ...) generally lie close to large streams with high discharge. The correlation with elevation might explain why initially the dependence of irrigation on discharge decreases. For large values




**Figure 6.** Partial dependence of the predictors and the dependent variable of (a) the classification random forests and (b) the regression random forests. The results for the ISEA3H grid choice is shown in magenta and the result for the longitude-latitude grid choice is shown in purple.

the large grid-size might be able to explain the differences between the grid, as for example along the Nile the irrigated areas

follow the river in a small band, being dispersed in the ISEA3H grid.





Lastly, we study the dependence of the GDP per capita categories on the probability to observe irrigation. We find a strictly positive relationship from the categories "Low", "Lower Middle", "Upper Middle" to "High". Therefore, the likelihood of croplands being irrigated is higher for areas with generally higher economic performance. Hence, adverse socio-economic conditions hinder the development of irrigated agriculture. This result complements the findings by Neumann et al. (2011),

who find similar effects considering government performance and government type as a potential factors explaining areas to become irrigated. The GDP per capita category "missing" corresponds to a relatively lower irrigation probability, which is in line the fact that with less areas being irrigated and less GDP per capita values observed in earlier time periods.

Panel b) of Figure 6 displays the partial dependence curves for the predictor variables of irrigation intensity, i.e. the amount of irrigation given a grid cell is irrigated. The most influential predictor, population density, positively impacts the amount of

irrigation.

Evaporation is also positively associated with the amount of irrigation, where the increase in irrigation appears to be almost linear in evaporation levels. The amount of irrigation negatively depends on precipitation levels, while discharge is positively correlated with irrigation intensity. Hence, the effect of water availability differs between different sources of water, where heavily precipitated areas seem to not require as much irrigation, while discharge might be used to feed irrigation systems.

The median potential productivity gain is positively associated with irrigation intensity, exhibiting a sharp peak in the dependence curve at the beginning of the distribution. Much of the tails is probably irrelevant for a real-world scenario, where irrigation would never happen in remote and dry regions, with a high potential for productivity increases from irrigation. Larger cell sizes in the ISEA3H grid mean "easier" access to streams (more area is in the same cell as the river), which is reflected in the higher plateau level.

Considering GDP per capita, we see irrigation intensity only slightly differing between the categories.

Assessing our results in the context of our hypotheses (see Table 1), we generally observe a consistent alignment between our empirical results and our previous theoretical consideration.

## 5    Conclusions

The careful choice of a discrete global grid system holds significant importance for conducting statistical analyses on a global

scale. In this paper, we make use of historical global irrigation data from the last century, to compare the standard longitude-latitude grid to the ISEA3H discrete global grid. We employ a stacked random forest framework to model likelihood of irrigation and irrigation intensity, once an area is irrigated, as a function of potential drivers. We identify population density and the potential productivity increase in terms of crop yield as the most influential factors for the decision to irrigate and population density and factors accounting for water availability as drivers for intense irrigation. We further point to GDP per capita as

having influence on irrigation behaviour.

Comparing the two grid choices, we find that the ISEA3H geodesic discrete global grid corresponds to a higher prediction accuracy. Using the assigned test data, the model built on the geodesic discrete global grid produces a 28% lower root



normalised mean squared prediction error compared to the model built on the longitude-latitude grid. This result is robust to different normalisation definitions.

In terms of the global irrigation prediction pattern, we find that the ISEA3H grid comes closer to the actually observed irrigation benchmark. While the longitude-latitude grid choice leads to some highly under-predicted areas in India, East-Asia and the United States, the ISEA3H grid choice is associated with under-prediction in almost the same areas, although much smaller in magnitude. Although the increase in predictive accuracy might partly be due to the fact that the change in grid cell structure changes the scale and therefore the range of values of the targeted irrigation variable, the advantages of the uniformly

structured ISEA3H grids are evident and should be explored and tested in future research.

While the combination of water availability, climate, and socioeconomic data offer valuable insights into the role of discrete global grid choice and the drivers of historical irrigation expansion, it is clear that our setting does not come without limitations. We, for example, neglected seasonality, meaning that yearly values were used for the analysis. However in reality water availability is much more relevant in the growing season than in the off season. We consider the discrete global grid

induced by the longitude-latitude graticule and the ISEA3H geodesic discrete global grid. While we offer new evidence about the potential accuracy increase using a geodesic discrete global grid, our methodology does not include an exhaustive search for the best-possible grid choice. Our goal is rather to set a starting reference point for future research designs.

We model irrigation fraction as a function of precipitation, discharge, evaporation, population density, potential productivity increase in terms of crop yield, and GDP per capita. While these are important drivers of irrigation, there are likely other

contributing factors that we are not able to capture in our analysis, such as the access to groundwater, irrigation subsidies, or other socioeconomic factors such as the type of government. The access to spatially explicit information would allow researchers to further explore these potential drivers.

Another interesting avenue for future research is to include time-lag in the analysis. Potentially, not the data of the year 1990 is most indicative for explaining the irrigation fraction of 1990, but for example the (average) information of the previous

decade.

Lastly, irrigation data and the data of predictors are based on a large variety of sources from different years, which have likely introduced uncertainties. However, the data sourced in our study comprise a selection of the most appropriate data available.

Acknowledging these limitations, we consider our analysis as an important step towards understanding the role of discrete global grids in global statistical modelling. Particularly, exploring the application of the ISEA3H geodesic grid system in

different global analytical contexts presents an intriguing avenue for future research.

*Code and data availability.* The code and data used in this study are publicly available for download at Zenodo https://doi.org/10.5281/zenodo.10012830.

**Appendix A**



**Figure A1.** Evolution of the global means of the predictor variables across the study period 1902 to 2005. a) Population density, b) discharge, c) evaporation, d) median increase in potential productivity, e) precipitation and f) GDP per capita.





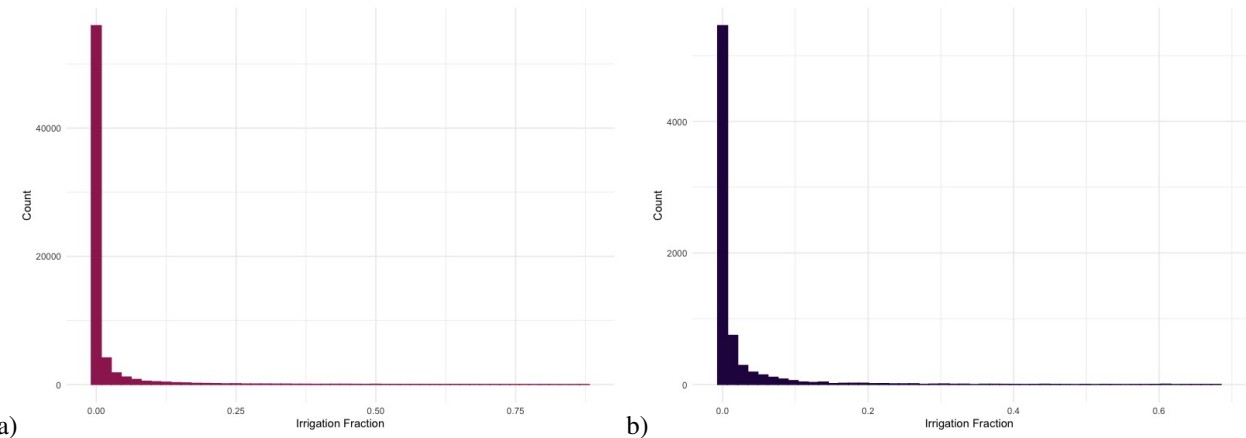

**Figure A2.** Histograms, showing the irrigation fraction on the $x$-axes and the corresponding frequency of the observational data used in the analysis in (a) the longitude-latitude grid and (b) the ISEA3H grid on the $y$-axes.





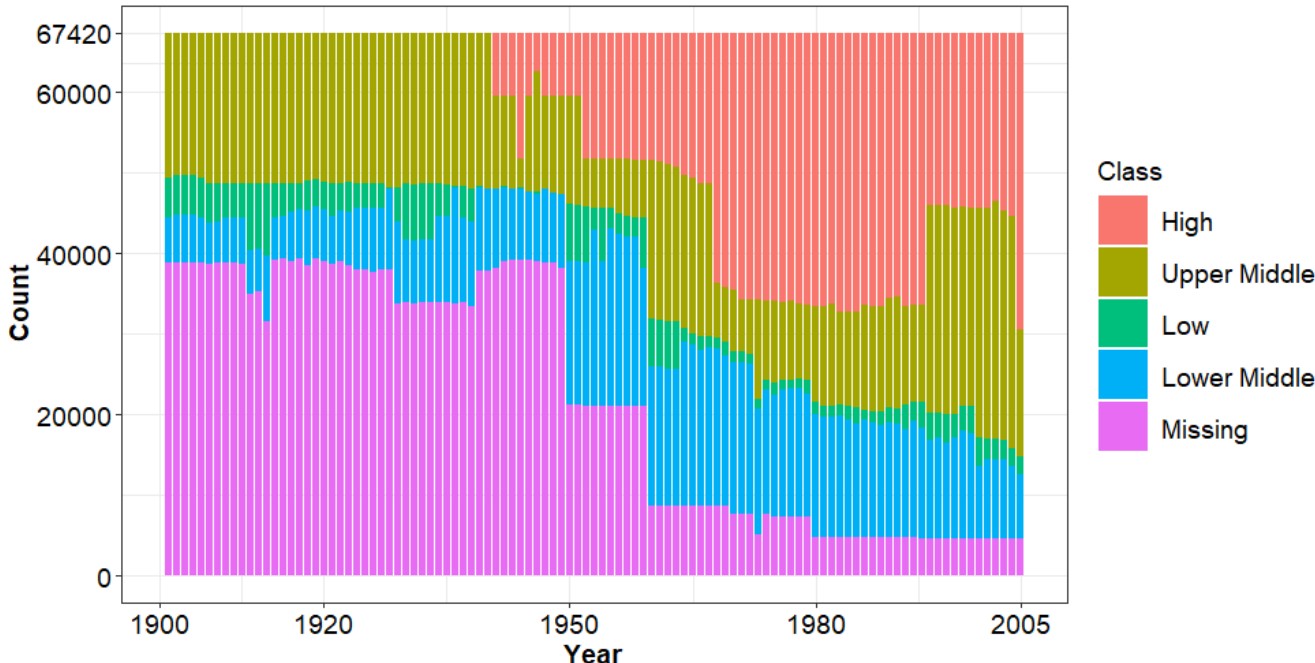

**Figure A3.** Frequency of GDP per capita categories over the study period 1902 to 2005.

a)

b)

**Figure A4.** Irrigation fraction in 2000 in a) the longitude-latitude discrete global grid and b) the ISEA3H discrete global grid. Irrigation fraction reflects the area irrigated of each grid cell and is based on the global Historical Irrigation Dataset (see Siebert et al., 2015).



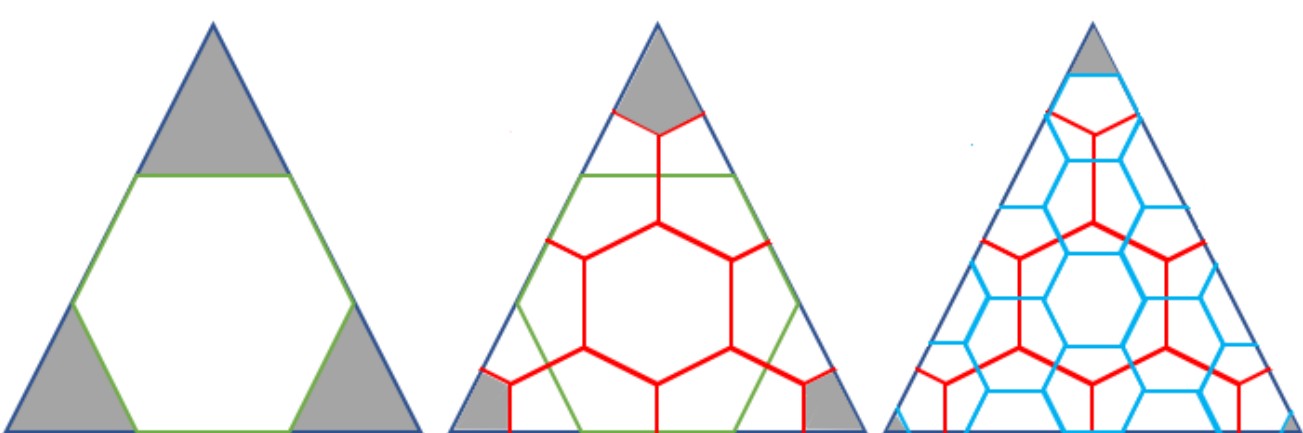

**Figure A5.** Recursive partitioning aperture 3 method. The hexagonal pattern is recursively constructed on top of the base icosahedron. The first resolution is illustrated by the green hexagon, directly constructed inside a triangular face of the base icosahedron. The construction of the resolution 2 grid is displayed in red in the middle. The resolution 3 hexagonal pattern is illustrated on the right side. Increasing the resolution by one, leads to hexagons with a size of one third of the original hexagon size. The grey left over areas are the reason why overall, a few pentagonal faces are needed to cover the Earth's surface. The image is based on an illustration by de Wiljes (2015).





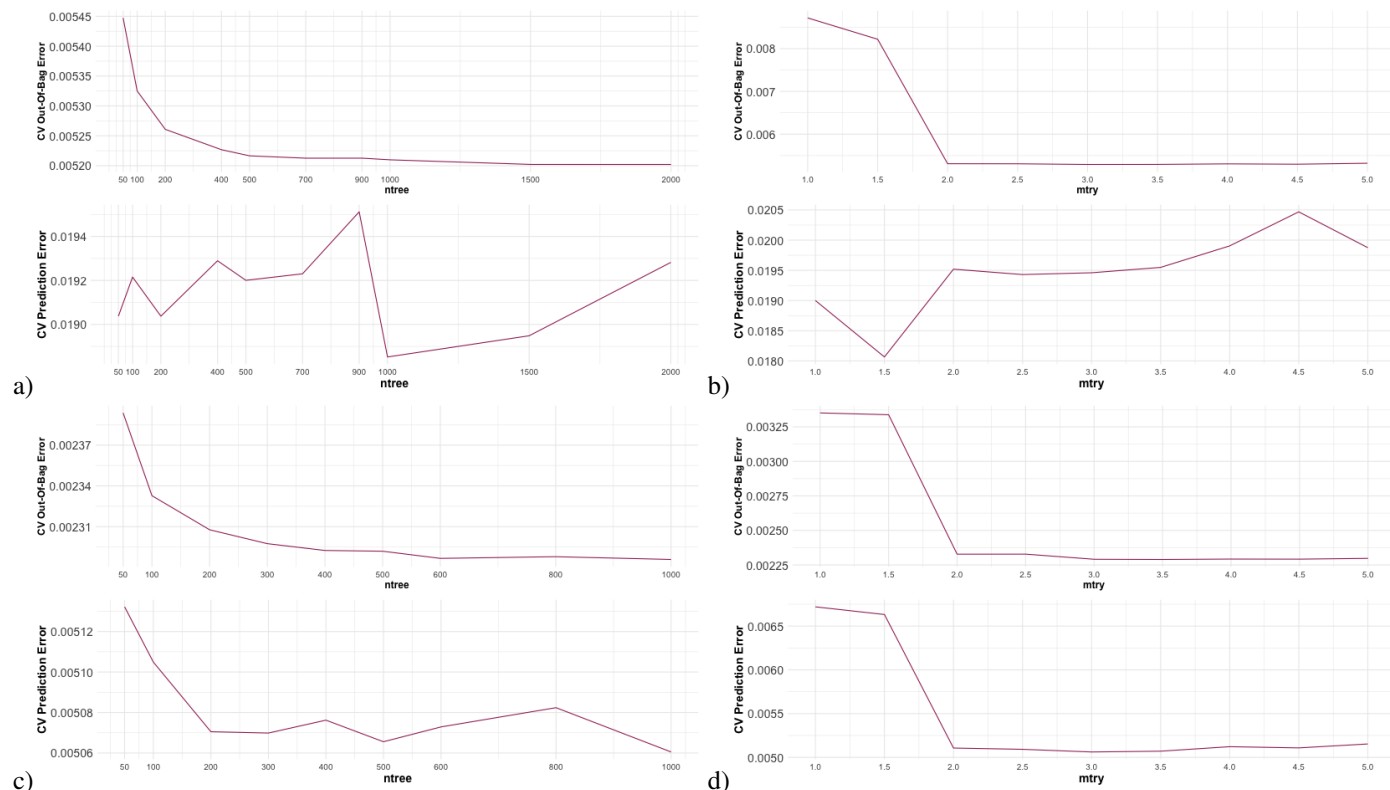

**Figure A6.** Cross-validation results of the longitude-latitude grid choice. The out-of-bags error and the prediction error are displayed as a function of changing hyperparameter values for a) ntree in the classification random forest, b) mtry in the classification random forest, c) ntree in the regression random forest and d) mtry in the regression random forest.





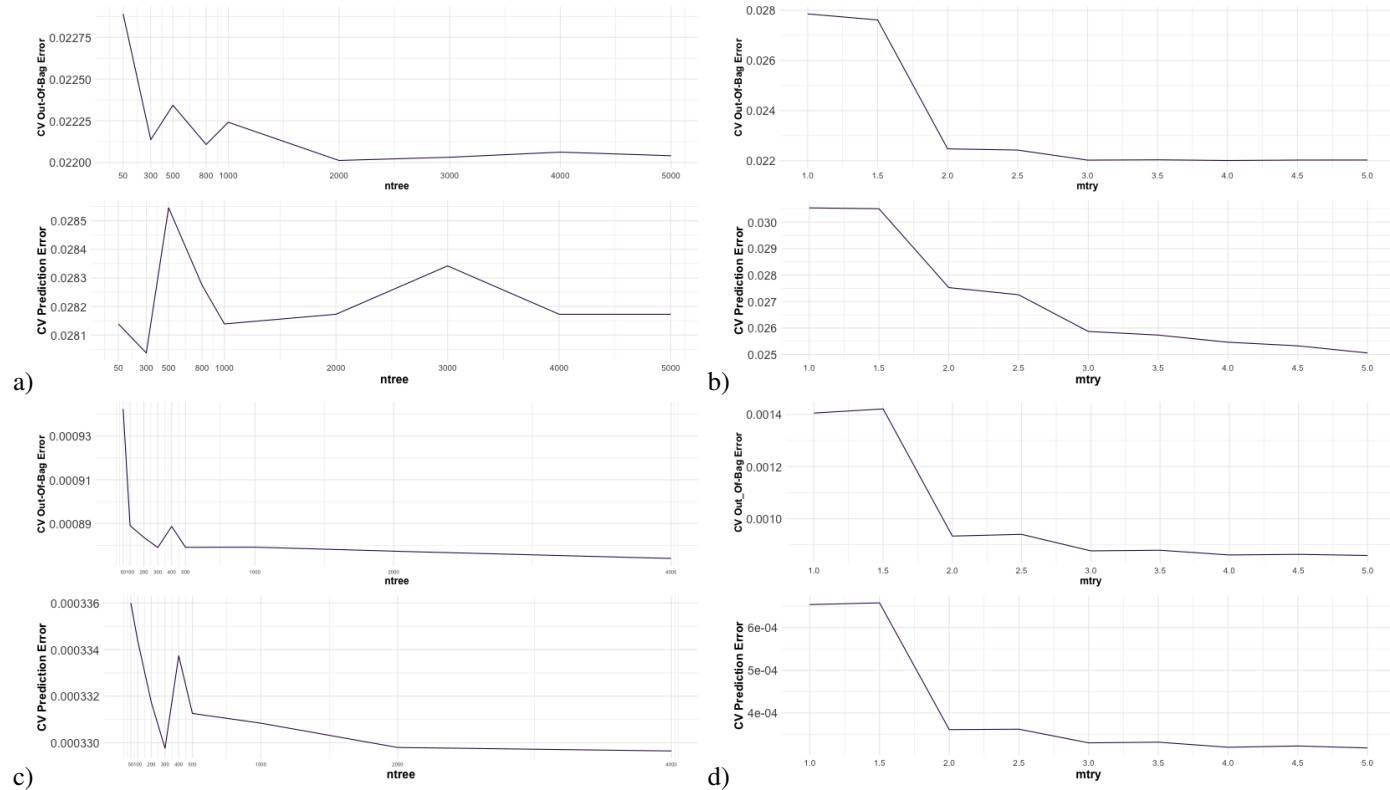

**Figure A7.** Cross-validation results of the geodesic discrete global grid choice. The out-of-bags error and the prediction error are displayed as a function of changing hyperparameter values for a) ntree in the classification random forest, b) mtry in the classification random forest, c) ntree in the regression random forest and d) mtry in the regression random forest.





**Table A1.** Summary statistics of the training data (1902-1999)

| | Mean | Standard deviation | Minimum | Maximum | Median |
|---|---|---|---|---|---|
| | (1) | (2) | (3) | (4) | (5) |
| **A. Longitude-latitude grid (n = 6.607,160)** | | | | | |
| Irrigation fraction | 0.0077 | 0.0375 | 0.0000 | 0.9220 | 0.0000 |
| Population density | 19.5986 | 72.4894 | 0.0000 | 9832.0000 | 1.0000 |
| Precipitation | 716.3860 | 712.2138 | 0.0000 | 11155.0000 | 478.9372 |
| Evaporation | 116.6513 | 80.7215 | 0.0000 | 953.9896 | 97.3343 |
| Discharge | 469.6246 | 4524.4351 | 0.0000 | 270078.8232 | 28.0981 |
| Median increase in productivity | 7.6580 | 65.9509 | -0.5596 | 17365.5508 | 0.0053 |
| **B. ISEA3H grid (n = 730,917)** | | | | | |
| Irrigation fraction | 0.0084 | 0.0318 | 0.0000 | 0.8077 | 0.0000 |
| Population density | 23.5708 | 69.3340 | 0.0000 | 4575.0000 | 2.0000 |
| Precipitation | 905.5131 | 848.5934 | 0.0000 | 10853.0000 | 609.2273 |
| Evaporation | 134.0022 | 84.9666 | 0.0000 | 715.0584 | 116.7027 |
| Discharge | 517.9301 | 3245.9758 | 0.0000 | 134255.6759 | 70.7079 |
| Median increase in productivity | 8.9070 | 53.7531 | -0.1054 | 4313.1124 | 0.0421 |

Notes: Panel A summarizes the descriptive statistics of the test data set in the original longitude-latitude grid. The test data set contains the years 1902 to 1999. Panel B summarizes the descriptive statistics of the ISEA3H grid, i.e. after transforming the data to the hexagonal grid. The GDP per capita predictor is excluded from this summary table, as it is a factor variable.



**Table A2.** GDP per capita category assignment

| Class | GDP Per Capita |
|---|---:|
| High | $\geq 12276\$$ |
| Upper Middle | $> 3975\$ - 12275\$$ |
| Lower Middle | $> 1005 - 3975\$$ |
| Low | $\leq 1005\$$ |
| Missing | $-$ |

Notes: GDP per capita classification by income level for the reference year 2011, based on the classification of the World Bank (2011).





**Table A3.** Pearson correlation coefficient

| | Population density | Median increase in productivity | Discharge | Precipitation |
|---|---|---|---|---|
| | | **Pearson Correlation Coefficient** | | |
| Median increase in productivity | −0.0212 | | | |
| Discharge | 0.0116 | −0.009 | | |
| Precipitation | 0.1349 | −0.094 | 0.1111 | |
| Evaporation | 0.2403 | −0.0417 | 0.0588 | 0.720 |

Notes: In this table, the correlation matrix of the Pearson correlation coefficient of the predictors is presented. The displayed values are the lower half of the correlation matrix.





**Table A4.** Variance inflation factor

| | Population density | Median increase in productivity | Discharge | Precipitation | Evaporation |
|---|---|---|---|---|---|
| | **Variance Inflation Factor** | | | | |
| VIF | 1.065230 | 1.010816 | 1.013481 | 2.124101 | 2.183165 |

Notes: This table displays the variance inflation factor (VIF) of the predictor variables. The measure is used to detect multicollinearity between potential predictor variables. A VIF below 5, means that the respective variable is not collinear to the other variables (James et al. (2013)).



*Author contributions.* SW, FS, TK and JdW conceptualized the underlying research and designed a choice of models and methodologies for
data analysis. SW performed the research and led the project. SW, FS, TK and JdW analyzed the results. SW wrote the original draft of the
manuscript. FS, TK and JdW reviewed and edited the manuscript.

*Competing interests.* The contact author has declared that none of the authors has any competing interests.

*Acknowledgements.* This research has been partially funded by Deutsche Forschungsgemeinschaft (DFG) - SFB1294/1 - 318763901. FS
acknowledges funding by the CE-Land+ project of the German Research Foundation's priority program SPP 1689 "Climate Engineering –
Risks, Challenges and Opportunities?" as well as by the Global Challenges Foundation via Future Earth.





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
