# Peer review of "Drivers of global irrigation expansion: the role of discrete global grid choice"

_Hydrology and Earth System Sciences, 2023_

## Author Response (AR1)

**HESS Review – Response**

We would like to thank all referees, members of the community and the editor for taking the time to read our manuscript and for the comments and suggestions. We summarized the comments below and provide a response for each.

**RC1**

This manuscript is dedicated to exploring the use of hexagons to represent irrigation maps and identify irrigation drivers, which is interesting. However, this study has significant flaws.

1. First, area deformation is more severe at high latitudes, but the global irrigated area is more distributed in the mid-latitudes and this effect would not be significant.
   **Reply:** It is true that most irrigation occurs in the mid-latitudes, where area deformation is less severe. However, we include all terrestrial grid cells to demonstrate that varying grid cell sizes significantly impact global statistical analysis. This is our main point, generalized from the irrigation application. In our initial step, we predict whether a grid cell is irrigated, considering both irrigated and rainfed grid cells equally important. Therefore, we cannot eliminate rainfed grid cells from our analysis.

2. Second, I think there are currently feasible ways to correct for the effect of latitudinal differences on area when counting global land averages.
   **Reply:** It is true that there are feasible methods to correct for latitudinal area distortion in statistical models. To address this, we included an additional model based on the longitude-latitude grid using classical area weights in the estimation and compared the results to our other models (section 3.3). Our findings indicate that the area weights do not significantly improve predictive accuracy, unlike our models based on equal-area grids (Table 2).

3. Third, the hexagon used is too large, too low resolution, and contains a large error. It is difficult to compare it directly with higher resolution grid data.
   **Reply:** We agree that our manuscript benefits from a sensitivity analysis with respect to grid cell resolution. We thus take a more systematic approach and include comparisons to the ISEA3H grids at resolutions 8 and 9, such that the total number of cells is in the same ballpark as the longitude-latitude grid, or that the cells are about the same size as the average grid cell size of the longitude-latitude grid (section 3.2, lines 187-197). We present the expanded results in Table 2 and section 4.1. This addresses the reviewers points by including versions that are comparable to the higher grid resolution data. We are thus able to assess whether results are dependent on changes in grid cell size. Further, we use bootstrapping to be able to compare the predictive accuracy between all the grid choices and see, whether the differences are of statistical significance. We find that classical area weights do not significantly improve the results of the original

longitude-latitude model. However, all the ISEA3H grid choices produce significantly lower error values compared to the longitude-latitude grid. This shows that our results are not dependent on the grid cell size of our equal-area ISESA3H grid (section 4.1, lines 285-295).

4. Fourth, the importance of the random forest variables is strongly dependent on the data resolution, and the importance of variables with low resolution may be underestimated.
**Reply:** We included the results of the variable importance for the two additional (finer) ISEA3H grid and for the area weighted longitude-latitude model (Figure 4 and section 4.2.1, lines 327-330). We find that the order of the importance of the predictors does not change with grid cell size. To even further validate this, we include the variable importance in the bootstrapping with 500 replications for each specification, and we see that the order is robust (section 4.1, 285-300).

5. Fifth, the sample sizes are an order of magnitude different when comparing precision in scatterplots, and the supposed improvement in precision derived in this way is not necessarily reliable.
**Reply:** We agree that this comparison alone is not informative enough. We add the same graphs for the additional, finer equal-area grids to complement the analysis (Figure 2). In fact, we see that the two grids with a similar number of grid cells (longitude-latitude and the ISEA3H at resolution 9) have a very similar scatterplot and therefore, the additional comparison in terms of global irrigation pattern and error values are necessary to draw conclusion about prediction accuracy (section 4.1 - lines 301-313).

6. Sixth, the writing in this manuscript could be improved. For example, it is not necessary to write the results and findings of the study in the last paragraph of the INTRODUCTION.
**Reply:** Thank you for this suggestion. We worked on condensing our manuscript and making the motivation more clear. We deleted and shortened multiple paragraphs (also removing the results from the end of the Introduction) and aimed to be more precise in our reasoning.

**CC1**

The study presents an interesting comparison of discrete grid choice in a statistical modelling context. However, I think that the study does not really provide a fair comparison between the square and hexagonal grids.

1. The main reason is that the two grids are of vastly different size with the hexagonal grid containing nearly 10 times fewer cells than the original square grid. The model trained on hexagonal grid is working on aggregates of the original data, while the square

grid model works with data in the original resolution. It is generally easier to model averaged data than data with original granularity. I would expect that the difference mostly disappears if the hexagonal grid has similar resolution than the 0.5 degree square grid. Relating to the above, the square grid would have to be approximately 1.5 degrees in resolution (7200 grid cells) to be approximately similar resolution to the hexagonal grid used. With this resolution, I do expect differences between the two configurations arising from the different shapes. However, I think that this is far too poor resolution to model global irrigation, given the availability of data and computing power available these days.

**Reply:** We appreciate the comment and agree that a sensitivity analysis regarding grid cell resolution enhances our manuscript. We have taken a more systematic approach by expanding our results table to include comparisons with ISEA3H grids at resolutions 8 and 9, ensuring that the total number of cells is comparable to the longitude-latitude grid (section 3.2, lines 187-197). Additionally, we present results using a classical area weighting approach in the longitude-latitude grid model to address the reviewer's concerns about resolution disparity.

Our findings indicate that while classical area weights do not significantly improve the results of the longitude-latitude model, all ISEA3H grid choices yield significantly lower error values, demonstrating that our results are not dependent on grid cell size (section 4.1, lines 285-295). Moreover, we included the results of variable importance for the two additional finer ISEA3H grids and for the area-weighted longitude-latitude model. The order of predictor importance does not change with grid cell size, and this order remains robust when validated through bootstrapping with 500 replications for each specification (section 4.1, 285-300).

By ensuring the ISEA3H grids have resolutions similar to the 0.5-degree square grid, we provide a fair comparison and address the reviewer's concern about the easier modeling of averaged data in larger hexagonal grids. This approach allows us to assess the true impact of grid shape and size on our modeling of global irrigation.

2. The use of the hexagonal grid is justified with the area distortion of the geographic coordinate systems, which the hexagonal grid in a geographic coordinate system. However, I would like to point out that the use of geographic coordinate system should have no influence in statistical modelling; computing correct areas for grids in geographical coordinate systems can be done with any major spatial library. If wrong areas are used, it is an error in the methodology, not a flaw in the geographic coordinate system. In addition, projecting the data to an equal-area projection (such as Mollweide or Equal Earth) eliminates the issue.

**Reply:** We appreciate your comment regarding the use of hexagonal grids and the potential for correcting area distortion in geographic coordinate systems. We acknowledge that accurate area computation can be achieved using major spatial libraries and

that projecting data to an equal-area projection, such as Mollweide or Equal Earth, can eliminate distortion issues.

However, our primary goal is to demonstrate that grid cell size variations significantly impact global statistical analysis. We also included models based on the longitude-latitude grid with classical area weights for comparison. Our findings indicate that the hexagonal grids consistently yield lower error values compared to the longitude-latitude grids, suggesting that the shape and distribution of grid cells also play a crucial role in the accuracy of our models. This underscores the importance of considering both grid shape and area distortion in spatial statistical analysis.

**RC2**

The manuscript "Drivers of global irrigation expansion: the role of discrete global grid choice" discusses the adoption of hexagonal cells, as an alternative to the traditional longitude-latitude grid for global statistical irrigation modeling. The authors suggest that the new method can mitigate the area distortion and increase predictive accuracy, and the random forest method was used to identify the potential drivers of historical global irrigation expansion. Overall, this study represents an attempt to address the challenges associated with traditional grid systems in global irrigation analysis. However, there are significant areas that require further investigation to fully substantiate the authors' claims, particularly the potential bias introduced by grid cell size, which may make the comparison less fair.

 Specific comments:

1. I think there seems to be a disconnect between the discussion of grid bias and the examination of global irrigation drivers in the Introduction. It would strengthen the manuscript if the authors explicitly linked how grid biases might affect the analysis of irrigation drivers, especially given the fact that significant irrigated areas are concentrated in low to mid latitudes where these biases are less pronounced (as shown in Fig. 1).
**Reply:** We acknowledge the importance of addressing grid biases in the analysis of global irrigation drivers. Our main point is to show that different grid cell sizes do matter in global statistical analysis, which we generalize from the irrigation application. We predict whether a grid cell is irrigated, considering both irrigated and rainfed cells as equally important, hence we cannot eliminate rainfed grid cells (cf. comment 1 by RC1). To strengthen our manuscript, we conducted a sensitivity analysis with respect to grid cell resolution. We expanded our results table to include comparisons with ISEA3H grids at resolutions 8 and 9, making the total number of cells comparable to the longitude-latitude grid. Additionally, we included a classical area weighting approach by incorporating weights in the longitude-latitude grid model (Table 2). This approach addresses the reviewers' concerns by providing versions that are comparable to higher grid resolution data (cf. comment 3 by RC1 and comment 1 by CC1).

Our findings show that classical area weights do not significantly improve the original longitude-latitude model results. However, all ISEA3H grid choices yield significantly lower error values compared to the longitude-latitude grid, indicating that our results are not dependent on the grid cell size of our equal-area ISEA3H grid (section 4.1, lines 285-295). This systematic approach allows us to link how grid biases might affect the analysis of irrigation drivers, even in regions where these biases are less pronounced.

2. For the data, LPJmL primarily uses biophysical inputs. If socioeconomic factors significantly influence irrigation practices, and are not included in the model's parameters, how can the simulation data be used to study the socioeconomic impact? Also, is there any validation of the LPJmL results with observational datasets?

   **Reply:** LPJmL uses historical inputs, which represent also socioeconomic influences: e.g. land use fractions (which crop has been grown where reflects e.g. farmers choices based on subsidies), management options such as irrigation systems etc (described in section 2 - lines 93-103). Our statistic analysis is designed to assess the influence of precisely these drivers.

3. The detailed description of the random forest algorithm (like Figure 2 and Algorithm 1) might be unnecessary if no modifications were made to the algorithm in this study, given the popularity of random forest in the geoscience literature.

   **Reply:** Thank you for this suggestion, we shortened this section and took out the graph explaining the random forest method.

4. For the results in Figure 3, given the difference in grid cell sizes between the longitude-latitude grid and the ISEA3H grid, I am concerned that the larger grid cells in the ISEA3H model may be smoothing out prediction errors that are apparent in the finer-scale longitude-latitude grid. Therefore, an analysis of how grid cell size might affect apparent accuracy would be appreciated. In addition, could you provide $R^2$ values for the predictive performance of each grid model, in addition to RMSE and NRMSE?

   **Reply:** We appreciate the comment and agree that a sensitivity analysis regarding grid cell resolution enhances our manuscript. We have expanded our results table to include comparisons with ISEA3H grids at resolutions 8 and 9, ensuring the total number of cells is comparable to the longitude-latitude grid (cf. our answer to comment 1). We also present results using a classical area weighting approach in the longitude-latitude model to address the reviewer's points (section 4.1, lines 285-295).

   Our analysis shows that while classical area weights do not significantly improve the longitude-latitude model, all ISEA3H grid choices yield significantly lower error values. This indicates that our results are not dependent on grid cell size. Additionally, we provide $R^2$ values in Table 2 for the predictive performance of each grid model, alongside RMSE and NRMSE, to give a more comprehensive view of the accuracy.

5. The same would apply to variable importance. Different grid sizes could lead to different interpretations of what is most important in determining irrigation expansion, potentially biasing the model results. In other words, the difference between the two grid systems in the study may be the result of different grid sizes. Therefore, it is worth investigating how the grid cell size may affect the importance of variables in the random forest model.

   **Reply:** We included the results of the variable importance for the two additional (finer) ISEA3H grid and for the area weighted longitude-latitude model (Figure 4 and section 4.2.1, lines 327-330). We find that the order of the importance of the predictors does not change with grid cell size (cf. comment 4 by RC1). To even further validate this, we include the variable importance in the bootstrapping with 500 replications for each specification, and we see that the order is robust (section 4.1, 285-300).

---

## Author Response (AR2)

**HESS Review – Response**

We would like to thank all referees and the editor for taking the time to assess our responses and for the helpful comments and suggestions. Below, we provide the response for each.

**Referee 1**

The authors present a work to use hexagonal cells for investigation of global irrigation expansion in comparison to that derived from using lat-lon based grid cells. This work is interesting for me. The authors find that the use of hexagonal cells can achieve a better prediction performance while being compared to the use of lat-lon grid cells based on the random forest method. My major concerns are why and how the hexagonal cells can work better than the lat-lon grid cells.

1. Line 115: Besides the '1' for fully irrigated and '0' for none irrigated, how about the cells partially irrigated?
   **Reply:** The variable is the irrigation fraction, so all values between 0 and 1 are possible. We rephrased the sentence to make it less ambiguous.

2. Figure 2: Why the resolution 9 (d) looks similar to the lon-lat (a)? Since the resolution 9 has a resolution finer than resolution 7 and 8, what I expected is that it will lead to a better performance than (a) and (b). Besides, the statistical performances can be displayed in the sub-figures.
   **Reply:** We describe this in section "Grid choice". Actually, the more aggregated the values are (larger grid cells) the better the prediction performance is. The resolution 9 grid on average has a similar grid cell size than the lat-lon grid, while the resolution 8 grid has a comparable total amount of cells. According to Figure 2, the predictions from the lat-lon grid and the resolution 9 grid perform similar with regards to a visual inspection of the scatter plot. Table 2, however, shows that the resolution 9 grid is still superior to the lat-lon grid, when analyzing the detailed statistics.

3. Line 27: Since the authors claimed the unique of northernmost regions, it will be interesting to find the difference of prediction in such areas based on the hexagonal and lat-lon cells. For example, can we find the differences between the high-latitude areas while being compared to that in the low-latitude areas, as the two regions have different geographic distortions while being mapped at grid cells.
   **Reply:** Please see also our reply to comment 4 below. With a uniform lat-lon grid, the cells in areas near the equator are too large compared to those in Europe. This leads to an unnatural number of samples in these grid cells, and the effect is amplified by the grid.

4. Section 4.2: The analysis is necessary, but I am more interested on the role of hexagonal cells in interpretation. Will the lat-lon grid cells lead to different explanations? The

more deeper discussions on this point, i.e., why and how the hexagonal cells can work better than the lat-lon grid cells, can help us better understanding the topic of this study.

**Reply:** We are not sure how to address this comment. Section 4.2 specifically compares the relative importance and partial dependencies between the lat-lon grid and the hexagonal grid. Since the grid cells are more uniform in shape, size, and area compared to a traditional lat-lon grid (with the exception of a very small number of cells), it is expected that this grid will perform better. This uniformity enhances the interpretability of the overall results and their significance for individual grid cells. The role of neighbouring cells becomes more pronounced, as the definition of a neighbour is now much clearer—each neighbour shares an equal-length border with the considered cell, which is not the case in the lat-lon grid.

**Referee 2 (Shijie Jiang):**

1. I appreciate the authors' comprehensive response to my previous concerns, which have been successfully addressed. I have only one minor editorial comment before the manuscript can be accepted: The bar plot colors in Figure 4 are too similar, making it difficult to distinguish between the bars. I recommend adjusting the colors to improve readability.

   **Reply:** We will do that, thank you!